# Dynamics of the Interaction Between Two Coherent States in a Cavity with Finite Temperature Decay

**Leonardi Hernández-Sánchez** , **Irán Ramos-Prieto** , **Francisco Soto-Eguibar** and **Héctor M. Moya-Cessa** *

Instituto Nacional de Astrofísica Óptica y Electrónica, Calle Luis Enrique Erro No. 1, Santa María Tonantzintla, Puebla 72840, Mexico; leo_nardi@inaoep.mx (L.H.-S.); iran@inaoep.mx (I.R.-P.); feguibar@inaoep.mx (F.S.-E.)
* Correspondence: hmmc@inaoep.mx

**Abstract:** In this study, we present an exact solution to the Lindblad master equation describing the interaction of two quantized electromagnetic fields in a decaying cavity coupled to a thermal reservoir at a finite temperature. The solution is obtained using the superoperator technique, leveraging commutation relations to factorize the exponential of the Lindblad superoperators into a product of exponentials. To demonstrate the applicability of this approach, we analyze the dynamics of the system both analytically and numerically for two initial conditions: nonentangled and entangled coherent states, exploring their temporal evolution. Additionally, we employ entropy and quantum discord analysis to characterize quantum correlations and analyze the behavior of entanglement (or lack thereof) during the evolution. This comprehensive analysis provides valuable insights into the behavior of open quantum systems and their interaction with the environment.

**Keywords:** Lindblad master equation; superoperators; quantum correlations

## 1. Introduction

The study of quantum systems in optical cavities has played a pivotal role in the advancement of quantum optics [1,2]. Optical cavities provide a controlled environment that enables exploration of nonclassical behaviors of quantized electromagnetic fields, making them ideal platforms for observing phenomena such as spontaneous emission, quantum decoherence, and system thermalization [3–6]. In the absence of dissipation, the field modes within these cavities remain isolated, and their dynamics can be accurately described using the Schrödinger equation [7–12]. However, when the cavity experiences decay and interacts with a thermal reservoir, the system undergoes irreversible processes, such as dissipation and decoherence, requiring the use of the Lindblad master equation to describe its time evolution [13–15].

Although the Lindblad master equation is a powerful tool for modeling open quantum systems, obtaining exact solutions is often an intractable task. However, recent advances have employed techniques based on superoperators and non-unitary transformations to derive exact solutions for systems comprising two decaying field modes, both at zero and finite temperatures [16,17]. These studies have shown that, at zero temperature, the dynamics of the system are dominated solely by cavity dissipation, while at finite temperatures, thermal excitations from the reservoir introduce significant modifications [17]. Furthermore, exact solutions are critical for benchmarking numerical methods and for exploring phenomena such as thermalization, decoherence, and entanglement dynamics in more complex systems [6,18,19].

Entanglement, in particular, stands out due to its fundamental role in quantum technologies such as quantum computing, cryptography, and communication [18,20–22]. Entangled states arise from the interaction of quantized fields and exhibit correlations that challenge classical explanations while also being highly susceptible to environmental interactions, which can induce quantum decoherence and even lead to the loss of entanglement as the system interacts with a thermal reservoir [4,19]. This sensitivity underscores the importance of understanding how entangled and unentangled states evolve under dissipative conditions, as well as how the system transitions between quantum and classical regimes [23,24]. Moreover, the study of the preservation or degradation of entanglement in open quantum systems is vital for the advancement of quantum technologies, where environmental interactions remain a significant challenge [14,25]. In addition to entanglement, quantum discord and classical correlations have emerged as essential tools for characterizing quantum correlations in open systems. Unlike entanglement, quantum discord captures broader quantum correlations that can persist even in separable states [26,27]. Studying quantum discord complements the analysis of entanglement, providing deeper insights into how correlations evolve and how systems transition between quantum and classical behaviors [28,29].

To contextualize this study within the larger literature, we note that previous work has focused primarily on specific cases of initial states, often neglecting a comparative analysis between entangled and unentangled scenarios [30–34]. Furthermore, while numerical approaches have been widely employed, there is a scarcity of analytical solutions that provide fundamental insights into the interaction between dissipation and thermal effects [6,13,35]. This gap highlights the need for a detailed theoretical and numerical exploration of these systems, addressing both entangled and nonentangled initial conditions.

This work aims to provide a detailed analysis of the Lindblad master equation describing the interaction of two quantized field modes in an optical cavity coupled to a thermal reservoir. Using the superoperator formalism, we explore how the initial states of the system, whether entangled or not, affect its temporal evolution under thermal dissipation. In Section 2, we introduce the theoretical framework that governs the physical system and the Lindblad equation that describes its dynamics. Section 3 presents the solution methodology, employing superoperator techniques to factorize the evolution operator through commutation relations. In Section 4, we analyze the evolution of two initially nonentangled coherent states, and in Section 5, we examine the evolution of two initially entangled coherent states. In Section 6, we perform a detailed analysis of entropy and quantum discord to study the behavior of quantum and classical correlations, providing insight into the dynamics of entanglement of the system. Finally, in Section 7, we present our conclusions.

## 2. Lindblad Master Equation

Consider the interaction between two quantized fields, denoted by $\hat{a}$ and $\hat{b}$, with each subject to Markovian decay processes with a common loss rate, $\gamma$. Both fields are coupled to a thermal reservoir characterized by an average thermal excitation number, $\bar{n}_{\text{th}} \geq 0$, as shown in Figure 1.

The Markovian dynamics of the reduced density matrix, $\hat{\rho}$, in the interaction picture is described by the Lindblad master equation [13–15]:

$$\frac{d\hat{\rho}}{dt} = -\,\mathrm{i}\,g\hat{S}\hat{\rho} + \mathcal{L}_a\hat{\rho} + \mathcal{L}_b\hat{\rho}, \tag{1}$$

where $\hat{S}\hat{\rho}$ is defined as $\hat{S}\hat{\rho} = [\hat{H}_{\text{int}}, \hat{\rho}]$, with $\hat{H}_{\text{int}} = \hat{a}\hat{b}^\dagger + \hat{a}^\dagger\hat{b}$ representing the interaction Hamiltonian, which describes the coherent coupling between the two bosonic field modes.

Here, $\hat{a}$ ($\hat{a}^\dagger$) and $\hat{b}$ ($\hat{b}^\dagger$) are the annihilation (creation) operators for each mode, and $g$ denotes the coupling strength between the two bosonic modes [7–12]. The terms $\mathcal{L}_c\hat{\rho}$, with $c = a, b$, represent the Lindblad superoperators—also known as the Lindbladian—for the decay processes of each field. The Lindbladian is defined as follows:

$$\mathcal{L}_c\hat{\rho} \equiv \gamma\left[(\bar{n}_{\text{th}} + 1)\left(2\hat{c}\hat{\rho}\hat{c}^\dagger - \hat{c}^\dagger\hat{c}\hat{\rho} - \hat{\rho}\hat{c}^\dagger\hat{c}\right) + \bar{n}_{\text{th}}\left(2\hat{c}^\dagger\hat{\rho}\hat{c} - \hat{c}\hat{c}^\dagger\hat{\rho} - \hat{\rho}\hat{c}\hat{c}^\dagger\right)\right]. \tag{2}$$

The Lindbladian describes how the bosonic system interacts with the thermal environment, capturing both the loss and gain of excitations. The first term accounts for the loss of excitations due to the interaction with the environment, where the factor $\bar{n}_{\text{th}} + 1$ represents the total number of excitations, including those induced by the thermal reservoir. This dissipation is modeled through the annihilation operators $\hat{c}$, with $\hat{c} = \hat{a}, \hat{b}$. The second term captures the gain of thermal excitations, with $\bar{n}_{\text{th}}$ representing the average number of excitations in equilibrium, and the creation operators $\hat{c}^\dagger$ model this gain. Together, these terms offer a comprehensive description of the dynamics of the system influenced by both the mutual interaction and the thermal environment (see Figure 1). It is also evident that at zero temperature, where $\bar{n}_{\text{th}} = 0$, the Equation (1) reduces to the case where only decay is present, as analyzed in [16].

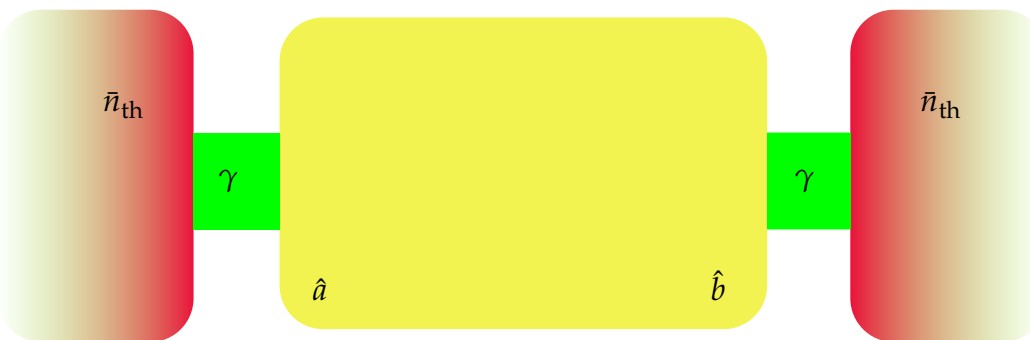

**Figure 1.** Schematic representation of two bosonic fields, $\hat{a}$ and $\hat{b}$, both coupled to the same thermal reservoir with an average thermal photon number $\bar{n}_{\text{th}}$. The interaction with the reservoir induces both loss and gain processes, characterized by dissipation mechanisms that include spontaneous emission, at a rate proportional to $\gamma(1 + \bar{n}_{\text{th}})$ and thermal absorption proportional to $\gamma\bar{n}_{\text{th}}$. These processes ensure that both fields experience decay at the same rate, $\gamma$, while maintaining equilibrium with the thermal reservoir, reflecting identical incoherent dynamics for each field.

## 3. Solution to the Lindblad Master Equation

In order to solve the Lindblad master Equation (1), it is important to note that, although the Lindblad superoperators individually do not commute with the interaction term, surprisingly, the sum of the Lindblad superoperators $\mathcal{L}_a\hat{\rho}$ and $\mathcal{L}_b\hat{\rho}$ commutes with the superoperator $\hat{S}\hat{\rho}$, that is,

$$\left[\mathcal{L}_a + \mathcal{L}_b,\ \hat{S}\right]\hat{\rho} = 0, \tag{3}$$

and, obviously, the Lindblad superoperators commute with each other. It is worth highlighting that the superoperators commute if and only if the decay rate is the same for both, i.e., $\gamma = \gamma_a = \gamma_b$, and the fields are coupled to two independent thermal reservoirs at the same temperature, which is characterized by an average number of thermal photons $\bar{n}_{\text{th}}$ (see Figure 1). From this, we can derive the formal solution to the Lindblad master Equation (1), given an initial condition $\hat{\rho}(0)$, as

$$\hat{\rho}(t) = e^{\mathcal{L}_a t}\, e^{\mathcal{L}_b t}\, e^{-\mathrm{i}gt\hat{S}}\, \hat{\rho}(0). \tag{4}$$

This solution requires applying the exponential functions of the superoperators to the initial condition in a specific way. This process is not straightforward, as each superoperator consists of a product or sum of operators. To apply these exponentials correctly, we must carefully decompose each superoperator. In the following, we will provide a detailed explanation of how to achieve this decomposition step by step.

### 3.1. Decomposition of the First Exponential

To efficiently apply the first exponential term, $e^{-\mathrm{i}gt\hat{S}}$, to an arbitrary initial condition $\hat{\rho}(0)$ as presented in (4), it is crucial to decompose the term into a more tractable form. For this purpose, we define the operators:

$$\hat{A} = \frac{\hat{a} + \hat{b}}{\sqrt{2}} \quad \text{and} \quad \hat{B} = \frac{\hat{a} - \hat{b}}{\sqrt{2}}, \tag{5}$$

which satisfy the relation

$$\hat{a}\hat{b}^{\dagger} + \hat{a}^{\dagger}\hat{b} = \hat{A}^{\dagger}\hat{A} - \hat{B}^{\dagger}\hat{B} = \hat{N}_A - \hat{N}_B, \tag{6}$$

where $\hat{N}_A = \hat{A}^{\dagger}\hat{A}$ and $\hat{N}_B = \hat{B}^{\dagger}\hat{B}$ represent the number of operators for the modes $\hat{A}$ and $\hat{B}$, respectively. Utilizing this result, the first exponential term in (4), when applied to $\hat{\rho}(0)$, corresponds to the unitary evolution driven by the interaction Hamiltonian and can be expressed as

$$e^{-\mathrm{i}gt\hat{S}}\hat{\rho}(0) = e^{-\mathrm{i}gt(\hat{N}_A - \hat{N}_B)} |\psi(0)\rangle \langle\psi(0)| e^{\mathrm{i}gt(\hat{N}_A - \hat{N}_B)}. \tag{7}$$

The unitary evolution captures the coherent dynamics dictated by the superoperator $\hat{S}$. The next step involves incorporating the nonunitary contributions arising from the Lindblad master equation. These terms account for dissipation and decoherence processes, ensuring a complete description of the open dynamics of the system.

### 3.2. Decomposition of the Second and Third Exponentials

The second and third exponential terms in (4), $e^{\mathcal{L}_c t}$ for $c = a, b$, govern the nonunitary dynamics of the modes $\hat{a}$ and $\hat{b}$, respectively. Although these terms play analogous roles, the main challenge lies in the explicit decomposition of the Lindbladian exponential $e^{\mathcal{L}_c t}$. To facilitate this decomposition, the Lindbladian superoperator $\mathcal{L}_c$, originally defined in (2), is rewritten in a more manageable form [17,36–38]:

$$\mathcal{L}_c \hat{\rho} = 2\gamma(\bar{n}_{\text{th}} + 1)\, \hat{c}\hat{\rho}\hat{c}^{\dagger} + 2\gamma\, \bar{n}_{\text{th}}\, \hat{c}^{\dagger}\hat{\rho}\hat{c} - \gamma(2\,\bar{n}_{\text{th}} + 1)\left(\hat{c}^{\dagger}\hat{c}\hat{\rho} + \hat{\rho}\hat{c}^{\dagger}\hat{c}\right) - 2\gamma\bar{n}_{\text{th}}\,\hat{\rho}$$

$$= \left(\hat{J}_{-}^{(c)} + \hat{J}_{+}^{(c)} + \hat{L}^{(c)} - 2\gamma\bar{n}_{\text{th}}\right)\hat{\rho}, \tag{8}$$

where we have utilized the commutation relation $\hat{c}\hat{c}^{\dagger} = \hat{c}^{\dagger}\hat{c} + 1$ and defined the following superoperators to represent the different dissipative contributions [17,37–39]:

$$\hat{J}_{-}^{(c)}\hat{\rho} = 2\gamma(\bar{n}_{\text{th}} + 1)\, \hat{c}\hat{\rho}\hat{c}^{\dagger}, \quad \hat{J}_{+}^{(c)}\hat{\rho} = 2\gamma\bar{n}_{\text{th}}\, \hat{c}^{\dagger}\hat{\rho}\hat{c}, \quad \hat{L}^{(c)}\hat{\rho} = -\gamma(2\,\bar{n}_{\text{th}} + 1)\left(\hat{c}^{\dagger}\hat{c}\hat{\rho} + \hat{\rho}\hat{c}^{\dagger}\hat{c}\right). \tag{9}$$

The decomposition highlights the distinct contributions to the dynamics of the system: $\hat{J}_{+}^{(c)}$ represents the photon gain process, $\hat{J}_{-}^{(c)}$ corresponds to photon loss, and $\hat{L}^{(c)}$ accounts for the dissipative effects arising from the interaction with the thermal reservoir. By reformulating the Lindbladian superoperator in this way, we can more efficiently analyze and

apply the nonunitary evolution of the system. This reformulation enables us to express the Lindbladian exponential as follows:

$$\hat{\rho}(t) = e^{-4\gamma\bar{n}_{\text{th}}t} \prod_{c=a,b} \left[ e^{\left(\hat{J}_{-}^{(c)}+\hat{J}_{+}^{(c)}+\hat{L}^{(c)}\right)t} \right] e^{-\mathrm{i}\,gt(\hat{N}_A-\hat{N}_B)} |\psi(0)\rangle \langle\psi(0)| e^{\mathrm{i}\,gt(\hat{N}_A-\hat{N}_B)}. \quad (10)$$

In this expression, the term $e^{-4\gamma\bar{n}_{\text{th}}t}$ represents a simple exponential decay factor, while the more complex dynamics is encapsulated in the exponential operator $e^{\left(\hat{J}_{-}^{(c)}+\hat{J}_{+}^{(c)}+\hat{L}^{(c)}\right)t}$.

It is straightforward to verify that these superoperators satisfy the commutation relations of the $SU(1,1)$ algebra, which is given by

$$\left[\hat{J}_{+}^{(c)}, \hat{J}_{-}^{(c)}\right]\hat{\rho} = 4\gamma^2\bar{n}_{\text{th}}(\bar{n}_{\text{th}}+1)\left[\frac{\hat{L}^{(c)}}{\gamma(2\bar{n}_{\text{th}}+1)} - 1\right]\hat{\rho}, \quad (11a)$$

$$\left[\hat{J}_{\pm}^{(c)}, \hat{L}^{(c)}\right]\hat{\rho} = \pm 2\gamma(2\bar{n}_{\text{th}}+1)\hat{J}_{\pm}^{(c)}\hat{\rho}. \quad (11b)$$

Given the commutation relations established, we propose an ansatz to factorize the sum of the superoperators in Equation (10) as a product of exponentials involving these superoperators. It is important to note that the order in which these superoperators are applied can be chosen freely. However, for convenience and to simplify the application of the solution to the initial conditions in which the field modes are in coherent states, it is advantageous to first apply powers of the annihilation operator, represented by the superoperator $\hat{J}_{-}^{(c)}$, followed by powers of the creation operator, represented by $\hat{J}_{+}^{(c)}$. This leads to the following ansatz:

$$\hat{\rho}(t) = e^{-4\gamma\bar{n}_{\text{th}}t}e^{2s(t)} \prod_{c=a,b} \left[ e^{r(t)\hat{J}_{+}^{(c)}} e^{q(t)\hat{L}^{(c)}} e^{p(t)\hat{J}_{-}^{(c)}} \right] e^{-\mathrm{i}\,gt(\hat{N}_A-\hat{N}_B)} |\psi(0)\rangle \langle\psi(0)| e^{\mathrm{i}\,gt(\hat{N}_A-\hat{N}_B)}. \quad (12)$$

By differentiating Equations (10) and (12) with respect to time, equating the resulting terms, and solving the resulting system of coupled differential equations, we derive the following solutions [38]:

$$p(t) = r(t) = \frac{1}{2\gamma\bar{n}_{\text{th}}}\frac{N(t)}{N(t)+1}, \quad (13a)$$

$$q(t) = \frac{1}{\gamma(2\bar{n}_{\text{th}}+1)}\{\gamma t + \ln[N(t)+1]\}, \quad (13b)$$

$$s(t) = 2\gamma\bar{n}_{\text{th}}t - \ln[N(t)+1], \quad (13c)$$

where $N(t) = \bar{n}_{\text{th}}(1 - e^{-2\gamma t})$.

Substituting these results into Equation (12) yields the final solution to the Lindblad master equation, as presented in Equation (4). This solution is now generalized and can be applied to any initial condition of the cavity, making it particularly valuable for investigating the dynamics of various quantum systems. This represents one of the key motivations that underpins the present manuscript.

## 4. Time Evolution of Nonentangled Coherent States

In this section, we analyze the temporal evolution of a system consisting of two field modes initially prepared in a nonentangled state described by [7–12]:

$$|\psi(0)\rangle = |\alpha\rangle_a \otimes |\beta\rangle_b. \quad (14)$$

This initial condition serves as the basis for studying how nonentangled coherent states evolve under the influence of the Lindblad dynamics, particularly focusing on the effects of dissipation and thermalization in the presence of a thermal reservoir.

### 4.1. Action of the First Exponential Operator

We begin by applying the first exponential operator, as expressed in Equation (7). Using the results obtained in (A4) and the property $e^{it\hat{n}}|\zeta\rangle = |\zeta e^{it}\rangle$ (where $\hat{n} = \hat{c}^\dagger\hat{c}$ is the number operator), we find that the transformed state evolves as

$$
e^{-igt\hat{S}}\hat{\rho}(0) = e^{-igt(\hat{N}_A - \hat{N}_B)}|\alpha\rangle_a\langle\alpha| \otimes |\beta\rangle_b\langle\beta| e^{igt(\hat{N}_A - \hat{N}_B)}
$$
$$
= \left|\frac{\alpha+\beta}{\sqrt{2}}e^{-igt}\right\rangle_A\left\langle\frac{\alpha+\beta}{\sqrt{2}}e^{-igt}\right| \otimes \left|\frac{\alpha-\beta}{\sqrt{2}}e^{igt}\right\rangle_B\left\langle\frac{\alpha-\beta}{\sqrt{2}}e^{igt}\right|. \tag{15}
$$

Using the transformation derived from Equation (A6), we map the evolved state back to the original field modes $\hat{a}$ and $\hat{b}$. This yields the following expression for the state after the application of the first exponential operator:

$$
e^{-igt\hat{S}}\hat{\rho}(0) = |\alpha'\rangle_a\langle\alpha'| \otimes |\beta'\rangle_b\langle\beta'|, \tag{16}
$$

where

$$
\alpha' = \frac{\alpha+\beta}{2}e^{-igt} + \frac{\alpha-\beta}{2}e^{igt}, \tag{17a}
$$
$$
\beta' = \frac{\alpha+\beta}{2}e^{-igt} - \frac{\alpha-\beta}{2}e^{igt}. \tag{17b}
$$

This result demonstrates how the initial nonentangled state evolves under the influence of the first exponential operator, effectively incorporating the interactions between the modes. The transformed coherent states, $|\alpha'\rangle_a$ and $|\beta'\rangle_b$, now reflect the effects of the interaction parameters and the temporal evolution while preserving the nonentangled structure of the system.

### 4.2. Action of the Second and Third Exponential Operators

As mentioned above, since the decay rate and the number of thermal excitations in the reservoir are the same for each mode in the cavity, the action of the exponential operators of the Lindblad superoperators $\mathcal{L}_a$ and $\mathcal{L}_b$ is practically identical. Therefore, it is sufficient to derive the results for a single mode (for example, for the operator $\hat{a}$) and then perform the tensor product with the other mode ($\hat{b}$).

Thus, we are now interested in showing how to apply the solution (12) to the previous result given by Equation (16). To do this, we will use the following results (for a more detailed derivation, the reader may refer to references [7,37,38]):

$$
e^{-2\gamma\bar{n}_{th}t}e^{s(t)} = \frac{1}{N(t)+1}, \tag{18a}
$$

$$
e^{p(t)\hat{J}_-^{(c)}}|\zeta'\rangle\langle\zeta'| = \exp\left[|\zeta'|^2\frac{N(t)}{N(t)+1}\frac{\bar{n}_{th}+1}{\bar{n}_{th}}\right]|\zeta'\rangle\langle\zeta'|, \tag{18b}
$$

$$
e^{q(t)\hat{L}^{(c)}}|\zeta'\rangle\langle\zeta'| = \exp\left[|\tilde{\zeta}(t)|^2 - |\zeta'|^2\right]|\tilde{\zeta}(t)\rangle\langle\tilde{\zeta}(t)|, \tag{18c}
$$

$$
e^{p(t)\hat{J}_+^{(c)}}|\tilde{\zeta}(t)\rangle\langle\tilde{\zeta}(t)| = \sum_{n=0}^\infty\frac{1}{n!}\left[\frac{N(t)}{N(t)+1}\right]^n\sum_{k=0}^n\sum_{m=0}^n\binom{n}{k}\binom{n}{m}
$$
$$
\times\sqrt{k!}\sqrt{m!}\,\tilde{\zeta}^*(t)^{n-k}\tilde{\zeta}(t)^{n-m}|\tilde{\zeta}(t),k\rangle\langle\tilde{\zeta}(t),m|, \tag{18d}
$$

where $\tilde{\zeta}(t) = \frac{\zeta' e^{-\gamma t}}{N(t)+1}$, $|\tilde{\zeta}(t),k\rangle = \hat{D}[\tilde{\zeta}(t)]|k\rangle$ is the displaced number operator [7–12], and $\zeta' = \alpha', \beta'$.

Thus, we can write the full solution to the Lindblad master Equation (1) as

$$
\hat{\rho}(t) = [\alpha]_a[\beta]_b\,|\tilde{\alpha}(t),k\rangle_a\langle\tilde{\alpha}(t),m| \otimes |\tilde{\beta}(t),k'\rangle\langle\tilde{\beta}(t),m'|, \tag{19}
$$

with

$$[\alpha]_a = \frac{\exp\left[|\tilde{\alpha}(t)|^2 - |\alpha'|^2\right] \exp\left[|\alpha'|^2 \frac{N(t)}{N(t)+1} \frac{\bar{n}_{\text{th}}+1}{\bar{n}_{\text{th}}}\right]}{N(t)+1} \sum_{n=0}^{\infty} \frac{1}{n!} \left[\frac{N(t)}{N(t)+1}\right]^n$$

$$\times \sum_{k=0}^{n} \sum_{m=0}^{n} \binom{n}{k} \binom{n}{m} \sqrt{k!} \sqrt{m!} \, \tilde{\alpha}^*(t)^{n-k} \, \tilde{\alpha}(t)^{n-m},$$

$$[\beta]_b = \frac{\exp\left[|\tilde{\beta}(t)|^2 - |\beta'|^2\right] \exp\left[|\beta'|^2 \frac{N(t)}{N(t)+1} \frac{\bar{n}_{\text{th}}+1}{\bar{n}_{\text{th}}}\right]}{N(t)+1} \sum_{n'=0}^{\infty} \frac{1}{n'!} \left[\frac{N(t)}{N(t)+1}\right]^{n'}$$

$$\times \sum_{k'=0}^{n'} \sum_{m'=0}^{n'} \binom{n'}{k'} \binom{n'}{m'} \sqrt{k'!} \sqrt{m'!} \, \tilde{\beta}^*(t)^{n'-k'} \, \tilde{\beta}(t)^{n'-m'}. \tag{20}$$

In this expression, $\tilde{\alpha}(t) = \frac{\alpha' e^{-\gamma t}}{N(t)+1}$, and $\tilde{\beta}(t) = \frac{\beta' e^{-\gamma t}}{N(t)+1}$. The terms $[\alpha]_a$ and $[\beta]_b$ represent prefactors that include the summation terms and components of the thermal distribution associated with each mode, $\hat{a}$ and $\hat{b}$, respectively. These terms account for the effects of dissipation and the number of excitations in each cavity owing to thermal fluctuations.

Note that as $t \to \infty$, $\tilde{\alpha}(t) \to 0$, $\tilde{\beta}(t) \to 0$, and $N(t) \to \bar{n}_{\text{th}}$. Therefore, we recover the thermal density matrix associated with the probability of finding $n$ photons in the mode $\hat{a}$ and $n'$ photons in the mode $\hat{b}$ (both at the same temperature $\bar{n}_{\text{th}}$), which is given by

$$\hat{\rho}(t \to \infty) = \frac{1}{(\bar{n}_{\text{th}}+1)^2} \sum_{n=0}^{\infty} \sum_{n'=0}^{\infty} \left(\frac{\bar{n}_{\text{th}}}{\bar{n}_{\text{th}}+1}\right)^{n+n'} |n\rangle_a \langle n| \otimes |n'\rangle_b \langle n'|. \tag{21}$$

## 5. Time Evolution of Entangled Coherent States

In the previous Section 4, it was shown that when the initial state consists of two coherent nonentangled states, the system evolves in such a way that the states remain separable throughout the evolution. The magnitudes of the coherent states change because of dissipation, but the states themselves do not become entangled. Naturally, one might wonder what happens if the initial state consists of two coherent entangled states [31,34]. To explore this, let us now consider an initial entangled state of the form:

$$|\psi(0)\rangle = \frac{1}{\mathcal{N}}(|\alpha\rangle_a \otimes |-\beta\rangle_b + |-\alpha\rangle_a \otimes |\beta\rangle_b), \tag{22}$$

where $\mathcal{N}$ is the normalization constant, which is given by

$$\mathcal{N} = \sqrt{2\left[1 + e^{-2(|\alpha|^2 + |\beta|^2)}\right]}. \tag{23}$$

Now, applying this initial condition to the first exponential operator (7) and using the same technique described in the previous section, we obtain

$$e^{-\mathrm{i}gt\hat{S}}\hat{\rho}(0) = \frac{1}{\mathcal{N}^2} \big(\, |\alpha'\rangle_a \langle\alpha'| \otimes |-\beta'\rangle_b \langle-\beta'| + |\alpha'\rangle_a \langle-\alpha'| \otimes |-\beta'\rangle_b \langle\beta'|$$

$$+ |-\alpha'\rangle_a \langle\alpha'| \otimes |\beta'\rangle_b \langle-\beta'| + |-\alpha'\rangle_a \langle-\alpha'| \otimes |\beta'\rangle_b \langle\beta'| \,\big), \tag{24}$$

where $\alpha'$ and $\beta'$ are identical to those obtained in (17).

It is important to note that, as in the previous case, the action of the exponential operators corresponding to the Lindblad superoperators $\mathcal{L}_a$ and $\mathcal{L}_b$ is essentially the same

and follows the same principles discussed above. This allows us to express the solution to the Lindblad master Equation (1) associated with the initial condition (22) as follows:

$$
\begin{aligned}
\hat{\rho}(t) = \frac{1}{\mathcal{N}^2} \big( & [\alpha_1]_a [\beta_1]_b \; |\tilde{\alpha}(t), k\rangle_a \langle \tilde{\alpha}(t), m| \otimes |-\tilde{\beta}(t), k'\rangle \langle -\tilde{\beta}(t), m'| \\
+ & [\alpha_2]_a [\beta_2]_b \; |\tilde{\alpha}(t), k\rangle_a \langle -\tilde{\alpha}(t), m| \otimes |-\tilde{\beta}(t), k'\rangle \langle \tilde{\beta}(t), m'| \\
+ & [\alpha_3]_a [\beta_3]_b \; |-\tilde{\alpha}(t), k\rangle_a \langle \tilde{\alpha}(t), m| \otimes |\tilde{\beta}(t), k'\rangle \langle -\tilde{\beta}(t), m'| \\
+ & [\alpha_4]_a [\beta_4]_b \; |-\tilde{\alpha}(t), k\rangle_a \langle -\tilde{\alpha}(t), m| \otimes |\tilde{\beta}(t), k'\rangle \langle \tilde{\beta}(t), m'| \big),
\end{aligned}
\tag{25}
$$

where, once again, $\tilde{\zeta}(t) = \frac{\zeta' e^{-\gamma t}}{N(t)+1}$, $|\tilde{\zeta}(t), k\rangle = \hat{D}[\tilde{\zeta}(t)] |k\rangle$ represents the displaced number operator, and $\zeta' = \alpha'$, $\beta'$. Additionally, the quantities $[\alpha_j]_a$ and $[\beta_j]_b$ (where $j = 1, 2, 3, 4$) are analogous to those obtained in (20), which can be explicitly found in Appendix B.

## 6. Numerical Results

The analytical results presented in Equations (19) and (25) describe the evolution of a system initially composed of two coherent states unentangled and entangled, respectively, inside a cavity with identical decay rates $\gamma$ and coupled to a thermal reservoir at temperature $\bar{n}_{\text{th}}$. In the case of the non-entangled initial condition, Equation (19) indicates that the system maintained its separable structure throughout evolution. Additionally, Equation (21) shows that, as $t \to \infty$, the system reached a stationary state, and the resulting density matrix reduced to the thermal matrix associated with the reservoir. This confirms that the nonentanglement between modes $\hat{a}$ and $\hat{b}$ was preserved throughout the evolution. On the other hand, for the entangled initial condition, Equation (25) describes a more complex dynamic. However, the analytical result alone does not allow for a direct conclusion on whether the entanglement was preserved or lost during the evolution of the system.

We can verify whether the system maintains its initial condition of nonentanglement or entanglement or, if these properties change during its evolution, if it is capable of analyzing the von Neumann entropy, which is a key tool for quantifying quantum correlations. In a bipartite system composed of modes $\hat{a}$ and $\hat{b}$, the entropy of a subsystem, such as mode $\hat{a}$, is defined as in [40]:

$$
S_a(t) = -\text{Tr}_a[\hat{\rho}_a(t) \ln \hat{\rho}_a(t)],
\tag{26}
$$

where $\hat{\rho}_a(t) = \text{Tr}_b[\hat{\rho}(t)]$ represents the reduced density matrix of mode $\hat{a}$, which is obtained by tracing out the contributions of mode $\hat{b}$. Similarly, the entropy for mode $\hat{b}$ is expressed as follows:

$$
S_b(t) = -\text{Tr}_b[\hat{\rho}_b(t) \ln \hat{\rho}_b(t)],
\tag{27}
$$

where $\hat{\rho}_b(t) = \text{Tr}_a[\hat{\rho}(t)]$.

For a separable state, the reduced density matrices $\hat{\rho}_a(t)$ and $\hat{\rho}_b(t)$ describe pure states. This implies that the entropy of both modes is zero ($S_a(t) = 0$ and $S_b(t) = 0$), reflecting the absence of correlations between them. Conversely, if the system generates entanglement, the reduced density matrices become mixed states, resulting in nonzero entropy. This analysis provides an effective means of identifying and characterizing quantum correlations within the system.

However, entropy alone does not distinguish whether the correlations in the system are classical or quantum. To address this, the analysis is complemented by quantum discord (QD), which quantifies the purely quantum correlations present in a state. Quantum discord is defined as in [26–29]:

$$
\mathcal{D}(\hat{\rho}) = \mathcal{I}(\hat{\rho}) - \mathcal{Q}(\hat{\rho}),
\tag{28}
$$

where $\mathcal{I}(\hat{\rho})$ is the quantum mutual information, which is a measure of the total correlations (both classical and quantum) present in the system. It is defined as

$$\mathcal{I}(\hat{\rho}) = S(\hat{\rho}_a) + S(\hat{\rho}_b) - S(\hat{\rho}), \tag{29}$$

where $S(\hat{\rho})$ represents the von Neumann entropy of the full system, and $S(\hat{\rho}_a)$ and $S(\hat{\rho}_b)$ are the entropies of the reduced subsystems $\hat{a}$ and $\hat{b}$, respectively. Quantum mutual information quantifies how much information is shared between subsystems, serving as a comprehensive measure of correlations. On the other hand, the term $\mathcal{Q}(\hat{\rho})$ represents the classical correlations (CC), and it is calculated as

$$\mathcal{Q}(\hat{\rho}) = \max_{\{\Pi_k\}} \left[ S(\hat{\rho}_b) - \sum_k p_k S(\hat{\rho}_b^k) \right], \tag{30}$$

where $\{\Pi_k\}$ denotes a set of projective measurements performed on subsystem $\hat{a}$, $p_k$ is the probability associated with the $k$-th measurement outcome, and $\hat{\rho}_b^k$ is the post-measurement state of subsystem $\hat{b}$. Classical correlations are determined by optimizing over all possible measurement bases on $\hat{a}$, reflecting the maximum extractable information of $\hat{b}$ based on measurements made on $\hat{a}$.

The combined analysis of entropy and quantum discord provides more detailed insight into the entanglement properties of the system. If $\mathcal{D}(\hat{\rho}) = 0$, it confirms the absence of quantum correlations, indicating that the system is not entangled, even if classical correlations may still be present. Conversely, a nonzero value of $\mathcal{D}(\hat{\rho})$ indicates the presence of quantum correlations, which may suggest entanglement, although it does not necessarily imply it. This distinction arises because quantum discord can also capture quantum correlations in separable states. Therefore, this approach allows us to distinguish between different types of correlations and analyze their evolution over time under the influence of dissipation and thermal coupling.

The following figures analyze the dynamics of the entanglement based on the initial conditions of the system. Figures 2 and 3 correspond to an initially nonentangled state at zero temperature ($\bar{n}_{\text{th}} = 0$) and finite temperature ($\bar{n}_{\text{th}} = 0.1$), respectively. Similarly, Figures 4 and 5 depict the case of an initial entangled state under the same temperature conditions.

Each figure is divided into four subfigures to provide a detailed analysis:

- Subfigures (I) and (II) show the entropies $S_a$ and $S_b$ of modes $\hat{a}$ and $\hat{b}$, respectively, as functions of time $t$ and decay rate $\gamma$.
- Subfigure (III) illustrates both entropies, $S_a$ (solid red line) and $S_b$ (dotted blue line), for a fixed decay rate $\gamma = 0.13$ as functions of time. Additionally, green dashed lines represent the entropy calculated from the reduced density matrix of the thermal state, allowing for comparison with the stationary entropy.
- The subfigure (IV) analyzes the quantum discord (QD, solid purple line) and the classical correlations (CCs, dotted black line) over time. This subfigure is key to drawing conclusions about the dynamics of entanglement and quantum correlations in the system.

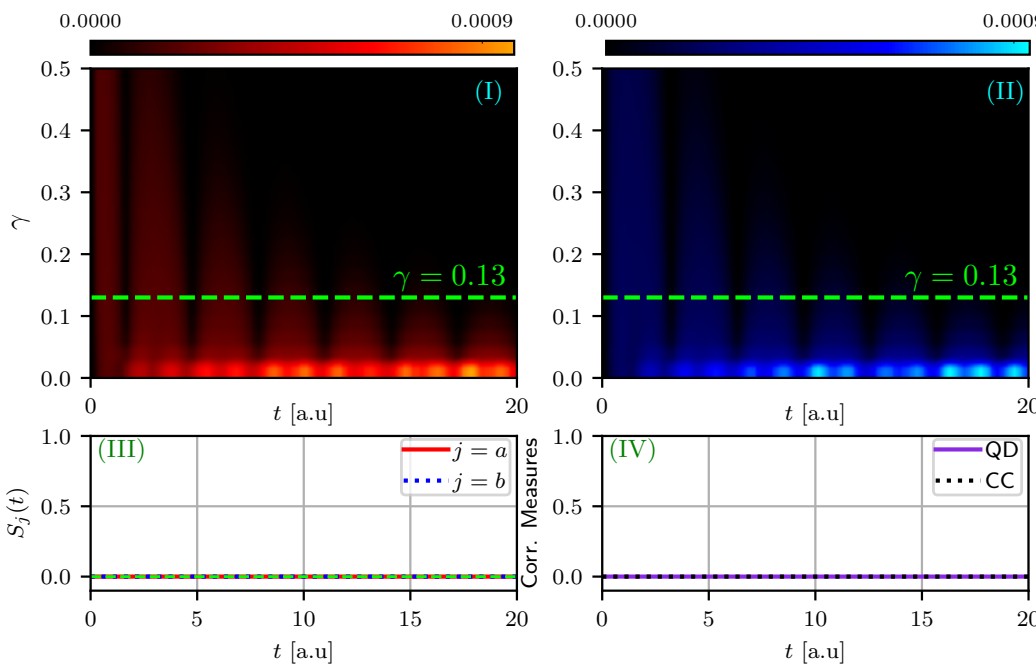

**Figure 2.** Analysis of the initially nonentangled state at zero temperature ($\bar{n}_{\text{th}} = 0$). Subfigures (**I**) and (**II**) show the entropies $S_a$ and $S_b$ of modes $\hat{a}$ and $\hat{b}$, respectively, as functions of time $t$ and decay rate $\gamma$. Subfigure (**III**) illustrates $S_a(t)$ (solid red line) and $S_b(t)$ (dotted blue line) for $\gamma = 0.13$, with the green dashed line representing the reduced entropy of the thermal state. Subfigure (**IV**) presents the quantum discord ($QD$, solid purple line) and classical correlations ($CC$, dotted black line) as functions of time. The parameters used are $\alpha = 2.5$, $\beta = 1$, and $g = 1$.

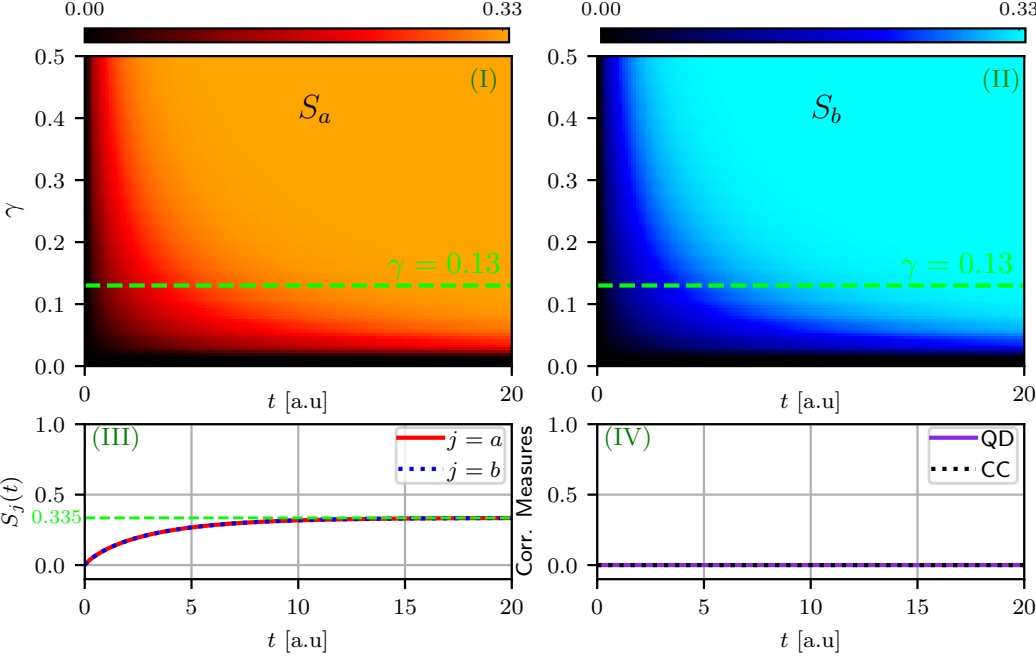

**Figure 3.** Analysis of the initially nonentangled state at finite temperature ($\bar{n}_{\text{th}} = 0.1$). Subfigures (**I**) and (**II**) show the entropies $S_a$ and $S_b$ of modes $\hat{a}$ and $\hat{b}$, respectively, as functions of time $t$ and decay rate $\gamma$. Subfigure (**III**) illustrates $S_a(t)$ (solid red line) and $S_b(t)$ (dotted blue line) for $\gamma = 0.13$, with the green dashed line showing the reduced thermal entropy. Subfigure (**IV**) presents the quantum discord ($QD$, solid purple line) and classical correlations ($CC$, dotted black line) over time. The parameters are the same as those used in Figure 2.

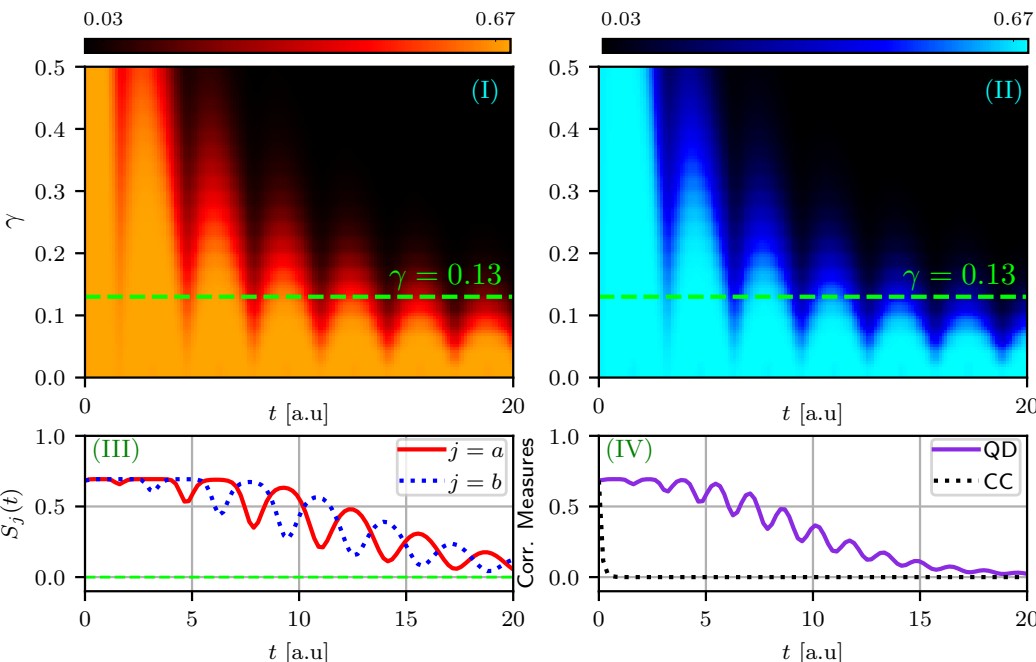

**Figure 4.** Analysis of the initially entangled state at zero temperature ($\bar{n}_{\text{th}} = 0$). Subfigures (**I**) and (**II**) show the entropies $S_a$ and $S_b$ of modes $\hat{a}$ and $\hat{b}$, respectively, as functions of time $t$ and decay rate $\gamma$. Subfigure (**III**) illustrates $S_a(t)$ (solid red line) and $S_b(t)$ (dotted blue line) for $\gamma = 0.13$, with the green dashed line representing the entropy calculated from the reduced density matrix of the thermal state. Subfigure (**IV**) presents the quantum discord (*QD*, solid purple line) and classical correlations (*CC*, dashed black line) as functions of time. The parameters are the same as those used in Figures 2 and 3.

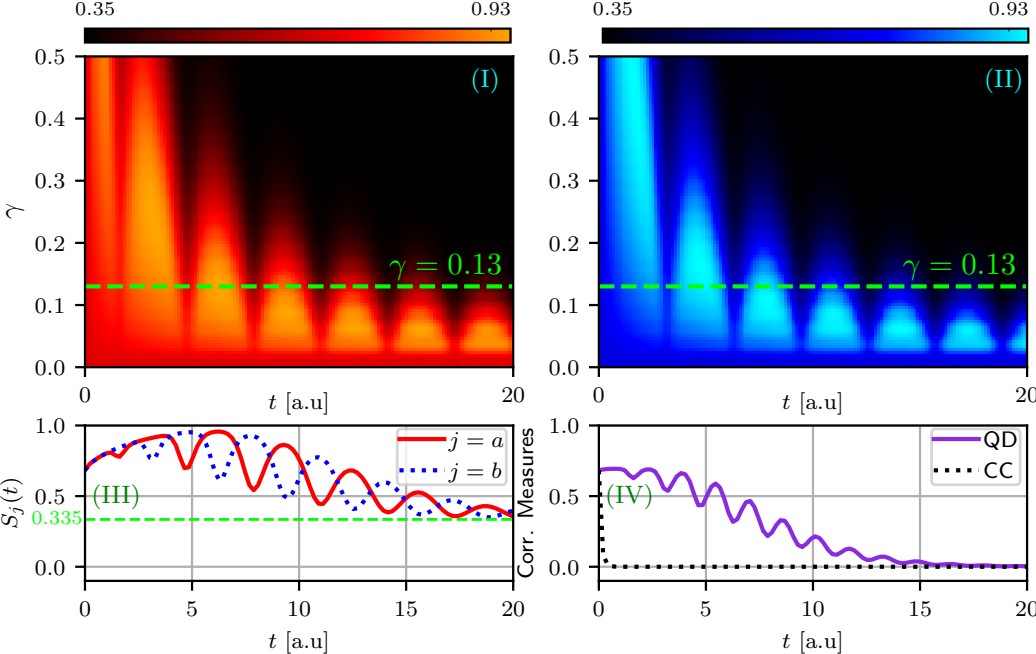

**Figure 5.** Analysis of the initially entangled state at finite temperature ($\bar{n}_{\text{th}} = 0.1$). Subfigures (**I**) and (**II**) show the entropies $S_a$ and $S_b$ of modes $\hat{a}$ and $\hat{b}$, respectively, as functions of time $t$ and decay rate $\gamma$. Subfigure (**III**) illustrates $S_a(t)$ (solid red line) and $S_b(t)$ (dotted blue line) for $\gamma = 0.13$, with the green dashed line representing the entropy calculated from the reduced density matrix of the thermal state. Subfigure (**IV**) presents the quantum discord (*QD*, solid purple line) and classical correlations (*CC*, dashed black line) as functions of time. The parameters are the same as those used in Figures 2–4.

The parameters used throughout the analysis are $\alpha = 2.5$, $\beta = 1$, and $g = 1$, ensuring consistency in all scenarios.

Detailed analysis of these cases leads to the following observations:

1. **Initially nonentangled state at zero temperature (Figure 2):** At zero temperature ($\bar{n}_{\text{th}} = 0$), the reservoir did not introduce thermal fluctuations, allowing the modes $\hat{a}$ and $\hat{b}$ to evolve independently while maintaining their purity. Consequently, the entropies $S_a(t)$ and $S_b(t)$ remained at zero throughout the evolution, as shown in subfigures (I) and (II), with any observed increases being numerical artifacts caused by the finite precision of the computational calculations. Subfigure (III) confirms that both entropies remained constant at zero for a fixed decay rate $\gamma = 0.13$, coinciding with the reduced and stationary entropy for this case. Furthermore, subfigure (IV) demonstrates that both the quantum discord (QD) and classical correlations (CCs) remained at zero, indicating the absence of any type of correlations. This analysis aligns with our analytical results, as the system preserved its initially nonentangled state, maintaining its separable structure and remaining in a pure state throughout the evolution.

2. **Initially nonentangled state at finite temperature (Figure 3):** At finite temperature ($\bar{n}_{\text{th}} = 0.1$), thermal fluctuations from the reservoir induced mixing in the initially pure states of modes $\hat{a}$ and $\hat{b}$, resulting in nonzero entropies $S_a(t)$ and $S_b(t)$ throughout the evolution. As shown in subfigures (I) and (II), these entropies increased over time due to interaction with the thermal reservoir and eventually converged to the same stationary value, $S_a = S_b = 0.335$. This stationary value, confirmed in subfigure (III) for a fixed decay rate $\gamma = 0.13$, corresponds to the reduced entropy associated with the thermal state, reflecting the equilibrium between the system and the reservoir. Subfigure (IV) illustrates that both the quantum discord (QD) and classical correlations (CCs) remained at zero throughout the evolution, indicating that no correlations, quantum or classical, were generated between the modes. The analysis confirms that the system retained its nonentangled nature throughout the evolution, with the increase in entropy attributed solely to thermal mixing induced by the reservoir, without the development of entanglement or quantum correlations. This behavior is consistent with the analytical results, demonstrating that the system evolved independently toward a thermal stationary state, where both modes exhibited equivalent properties, as reflected in the identical stationary entropies.

3. **Initially entangled state at zero temperature (Figure 4):** At zero temperature ($\bar{n}_{\text{th}} = 0$), the dynamics started with two initially entangled coherent states inside a decaying cavity. Subfigures (I) and (II) show that the entropies $S_a(t)$ and $S_b(t)$ were initially nonzero, reflecting the mixed nature of the reduced states due to the initial entanglement. During evolution, these entropies oscillated as a result of the internal exchange of quantum information between modes $\hat{a}$ and $\hat{b}$ and eventually decayed to zero, as confirmed in subfigure (III) for a fixed decay rate $\gamma = 0.13$, where the stationary entropies coincided with the reduced entropy of the thermal state, which was also zero. Subfigure (IV) reveals that the quantum discord (QD), initially nonzero due to the quantum correlations of the entangled state, oscillated and gradually decayed to zero, while the classical correlations (CCs) decayed rapidly to zero and remained there throughout the evolution, reflecting the absence of thermal fluctuations. Although the initial entanglement partially persisted while the entropies and discord oscillate, these oscillations indicate a progressive reduction in quantum correlations. Eventually, when the entropies and discord reached zero, the system completely lost its entanglement, evolving into a stationary state where both modes were decoupled, uncorrelated, and in pure individual states.

4. **Initially entangled state at finite temperature (Figure 5):** At finite temperature ($\bar{n}_{\text{th}} = 0.1$), the dynamics began with two initially entangled coherent states inside a decaying cavity. Subfigures (I) and (II) show that the entropies $S_a(t)$ and $S_b(t)$ remained nonzero throughout the evolution, exhibiting notable oscillations and higher values compared to the zero-temperature case. This behavior arose from the initial entanglement and the mixing induced by thermal fluctuations from the reservoir. Subfigure (III) demonstrates that both entropies, initially equal and nonzero, increased to a maximum due to thermal fluctuations and then decayed with oscillations toward the stationary value $S_a = S_b = 0.335$, making them consistent with the reduced thermal entropy. The coincidence of the stationary value between the initially entangled and nonentangled cases can be explained by the fact that the thermal reservoir imposed a statistical mixture independent of the system's initial configuration. Subfigure (IV) shows that the quantum discord (QD, solid purple line) started with a nonzero value, reflecting the initial quantum correlations, but oscillated and decayed more rapidly toward zero compared to the zero-temperature case due to thermal fluctuations. Meanwhile, the classical correlations (CCs, dashed black line) decayed quickly to zero and remained negligible throughout the evolution. While the initial entanglement partially persisted as long as the entropies and discord oscillated, these oscillations reflect a competition between the initial quantum correlations and the mixing effects induced by the thermal reservoir. During this phase, the system retained some degree of quantum correlation, but these correlations progressively weakened due to the dynamic exchange of information between the modes and the reservoir. Eventually, when the entropies and discord reached their stationary values, the entanglement was fully lost, and the system evolved into a thermal stationary state where both modes were decoupled, uncorrelated, and exhibited identical entropy values.

## 7. Conclusions

We have derived an exact solution to the Lindblad master equation that describes the interaction of two electromagnetic field modes in a decaying cavity coupled to a thermal reservoir at finite temperature.

Our results indicate that, for initial unentangled states, the system retains its separable structure throughout evolution, without generating quantum entanglement. At finite temperature, thermal fluctuations increase the entropy of the modes as a result of the statistical mixing induced by the reservoir. However, the quantum discord remains at zero, indicating that the correlations generated are not of a quantum nature.

For initially entangled states, the entanglement progressively weakens due to dissipation and thermal effects. At zero temperature, the entropies of the modes initially oscillate due to the exchange of quantum information within the system but eventually decay to zero, reflecting the transition to pure individual states and the total loss of entanglement. At finite temperature, the entropy oscillations are more pronounced, and the stationary state exhibits a nonzero entropy value corresponding to the thermal mixing imposed by the reservoir. In this case, the quantum discord also vanishes over time, signifying the complete loss of the initial quantum correlations.

These results highlight how environmental interactions, through dissipation and thermal fluctuations, destroy the initial quantum correlations of the system. For initially entangled states, the disappearance of quantum discord reflects the transition to a regime where the correlations are solely dictated by thermal equilibrium and lack any quantum character. This emphasizes the fragility of entanglement with environmental influences and underscores the practical limitations in preserving entanglement in open quantum systems.

**Author Contributions:** Conceptualization, F.S.-E. and H.M.M.-C.; Methodology, L.H.-S.; Investigation, L.H.-S., I.R.-P., F.S.-E., and H.M.M.-C.; Writing—original draft, L.H.-S.; Writing—review and editing, I.R.-P., F.S.-E. and H.M.M.-C.; Supervision, I.R.-P. All authors have read and agreed to the published version of the manuscript.

**Funding:** L. Hernández-Sánchez acknowledges the Instituto Nacional de Astrofísica, Óptica y Electrónica (INAOE), for the collaboration scholarship granted and the Consejo Nacional de Humanidades, Ciencias y Tecnologías (CONAHCYT), for the SNI Level III assistantship (CVU No. 736710).

**Institutional Review Board Statement:** Not applicable.

**Informed Consent Statement:** Not applicable.

**Data Availability Statement:** The data presented in this study are available upon request from the corresponding author.

**Conflicts of Interest:** The authors declare no conflicts of interest.

## Appendix A. Properties and Representation of Nonentangled Coherent States

It is well-known that the Glauber displacement operator [41,42] defines coherent states as

$$|\zeta\rangle = \hat{D}(\zeta)|0\rangle, \tag{A1}$$

where $\hat{D}(\zeta) = e^{\zeta \hat{a}^{\dagger} - \zeta^{*} \hat{a}}$, and $|0\rangle$ represents the vacuum state [7–12].

A state composed of two independent coherent states, $|\alpha\rangle$ (associated with mode $\hat{a}$) and $|\beta\rangle$ (associated with mode $\hat{b}$), can be written as a nonentangled state of the following form [43]:

$$|\psi\rangle = |\alpha\rangle_a \otimes |\beta\rangle_b = \hat{D}_a(\alpha)\hat{D}_b(\beta)|0\rangle_a \otimes |0\rangle_b. \tag{A2}$$

Using the definitions for the operators $\hat{A}$ and $\hat{B}$, introduced in Equation (5), the product of the displacement operators can be rewritten as

$$
\begin{aligned}
\hat{D}_a(\alpha)\hat{D}_b(\beta) &= \exp\left[\left(\alpha\hat{a}^{\dagger} - \alpha^{*}\hat{a}\right) + \left(\beta\hat{b}^{\dagger} - \beta^{*}\hat{b}\right)\right] \\
&= \exp\left[\left(\frac{\alpha+\beta}{\sqrt{2}}\hat{A}^{\dagger} - \frac{\alpha^{*}+\beta^{*}}{\sqrt{2}}\hat{A}\right) + \left(\frac{\alpha-\beta}{\sqrt{2}}\hat{B}^{\dagger} - \frac{\alpha^{*}-\beta^{*}}{\sqrt{2}}\hat{B}\right)\right] \\
&= \hat{D}_A\left(\frac{\alpha+\beta}{\sqrt{2}}\right)\hat{D}_B\left(\frac{\alpha-\beta}{\sqrt{2}}\right).
\end{aligned}
\tag{A3}
$$

Thus, the state in Equation (A2) can be expressed as follows:

$$|\psi\rangle = \left|\frac{\alpha+\beta}{\sqrt{2}}\right\rangle_A \otimes \left|\frac{\alpha-\beta}{\sqrt{2}}\right\rangle_B. \tag{A4}$$

Similarly, if we introduce a relative phase between the components of the state, it becomes

$$|\psi\rangle = \left|\frac{\alpha+\beta}{\sqrt{2}}e^{-i\eta}\right\rangle_A \otimes \left|\frac{\alpha-\beta}{\sqrt{2}}e^{i\eta}\right\rangle_B, \tag{A5}$$

which can be rewritten in terms of the original modes $\hat{a}$ and $\hat{b}$ as follows:

$$|\psi\rangle = \left|\frac{\alpha+\beta}{2}e^{-i\eta} + \frac{\alpha-\beta}{2}e^{i\eta}\right\rangle_a \otimes \left|\frac{\alpha+\beta}{2}e^{-i\eta} - \frac{\alpha-\beta}{2}e^{i\eta}\right\rangle_b. \tag{A6}$$

## Appendix B. Explicit Expressions for the Coefficients $[\alpha_j]_a$ and $[\beta_j]_b$

In this Appendix, we present the explicit expressions for the coefficients $[\alpha_j]_a$ and $[\beta_j]_b$ mentioned in the Equation (25). These coefficients are used to describe the time-dependent behavior of the subsystems and are derived based on the formalism introduced earlier. Their detailed forms are

$$
[\alpha_1]_a = \frac{\exp\left[|\tilde{\alpha}(t)|^2 - |\alpha'|^2\right]\exp\left[|\alpha'|^2\frac{N(t)}{N(t)+1}\frac{\bar{n}_{\text{th}}+1}{\bar{n}_{\text{th}}}\right]}{N(t)+1}\sum_{n=0}^{\infty}\frac{1}{n!}\left[\frac{N(t)}{N(t)+1}\right]^n
$$
$$
\times\sum_{k=0}^{n}\sum_{m=0}^{n}\binom{n}{k}\binom{n}{m}\sqrt{k!}\sqrt{m!}\,\tilde{\alpha}^*(t)^{n-k}\tilde{\alpha}(t)^{n-m},
\tag{A7a}
$$

$$
[\alpha_2]_a = \frac{\exp\left[-\left(|\tilde{\alpha}(t)|^2 - |\alpha'|^2\right)\right]\exp\left[-|\alpha'|^2\frac{N(t)}{N(t)+1}\frac{\bar{n}_{\text{th}}+1}{\bar{n}_{\text{th}}}\right]}{N(t)+1}\sum_{n=0}^{\infty}\frac{1}{n!}\left[\frac{N(t)}{N(t)+1}\right]^n
$$
$$
\times\sum_{k=0}^{n}\sum_{m=0}^{n}\binom{n}{k}\binom{n}{m}\sqrt{k!}\sqrt{m!}\,[-\tilde{\alpha}^*(t)]^{n-k}\tilde{\alpha}(t)^{n-m},
\tag{A7b}
$$

$$
[\alpha_3]_a = \frac{\exp\left[-\left(|\tilde{\alpha}(t)|^2 - |\alpha'|^2\right)\right]\exp\left[-|\alpha'|^2\frac{N(t)}{N(t)+1}\frac{\bar{n}_{\text{th}}+1}{\bar{n}_{\text{th}}}\right]}{N(t)+1}\sum_{n=0}^{\infty}\frac{1}{n!}\left[\frac{N(t)}{N(t)+1}\right]^n
$$
$$
\times\sum_{k=0}^{n}\sum_{m=0}^{n}\binom{n}{k}\binom{n}{m}\sqrt{k!}\sqrt{m!}\,\tilde{\alpha}^*(t)^{n-k}[-\tilde{\alpha}(t)]^{n-m},
\tag{A7c}
$$

$$
[\alpha_4]_a = \frac{\exp\left[|\tilde{\alpha}(t)|^2 - |\alpha'|^2\right]\exp\left[|\alpha'|^2\frac{N(t)}{N(t)+1}\frac{\bar{n}_{\text{th}}+1}{\bar{n}_{\text{th}}}\right]}{N(t)+1}\sum_{n=0}^{\infty}\frac{1}{n!}\left[\frac{N(t)}{N(t)+1}\right]^n
$$
$$
\times\sum_{k=0}^{n}\sum_{m=0}^{n}\binom{n}{k}\binom{n}{m}\sqrt{k!}\sqrt{m!}\,[-\tilde{\alpha}^*(t)]^{n-k}[-\tilde{\alpha}(t)]^{n-m},
\tag{A7d}
$$

and

$$
[\beta_1]_b = \frac{\exp\left[|\tilde{\beta}(t)|^2 - |\beta'|^2\right]\exp\left[|\beta'|^2\frac{N(t)}{N(t)+1}\frac{\bar{n}_{\text{th}}+1}{\bar{n}_{\text{th}}}\right]}{N(t)+1}\sum_{n'=0}^{\infty}\frac{1}{n'!}\left[\frac{N(t)}{N(t)+1}\right]^{n'}
$$
$$
\times\sum_{k'=0}^{n'}\sum_{m'=0}^{n'}\binom{n'}{k'}\binom{n'}{m'}\sqrt{k'!}\sqrt{m'!}\,[-\tilde{\beta}^*(t)]^{n'-k'}[-\tilde{\beta}(t)]^{n'-m'},
\tag{A8a}
$$

$$
[\beta_2]_b = \frac{\exp\left[-\left(|\tilde{\beta}(t)|^2 - |\beta'|^2\right)\right]\exp\left[-|\beta'|^2\frac{N(t)}{N(t)+1}\frac{\bar{n}_{\text{th}}+1}{\bar{n}_{\text{th}}}\right]}{N(t)+1}\sum_{n'=0}^{\infty}\frac{1}{n'!}\left[\frac{N(t)}{N(t)+1}\right]^{n'}
$$
$$
\times\sum_{k'=0}^{n'}\sum_{m'=0}^{n'}\binom{n'}{k'}\binom{n'}{m'}\sqrt{k'!}\sqrt{m'!}\,\tilde{\beta}^*(t)^{n'-k'}[-\tilde{\beta}(t)]^{n'-m'},
\tag{A8b}
$$

$$
[\beta_3]_b = \frac{\exp\left[-\left(|\tilde{\beta}(t)|^2 - |\beta'|^2\right)\right]\exp\left[-|\beta'|^2\frac{N(t)}{N(t)+1}\frac{\bar{n}_{\text{th}}+1}{\bar{n}_{\text{th}}}\right]}{N(t)+1}\sum_{n'=0}^{\infty}\frac{1}{n'!}\left[\frac{N(t)}{N(t)+1}\right]^{n'}
$$
$$
\times\sum_{k'=0}^{n'}\sum_{m'=0}^{n'}\binom{n'}{k'}\binom{n'}{m'}\sqrt{k'!}\sqrt{m'!}\,[-\tilde{\beta}^*(t)]^{n'-k'}\tilde{\beta}(t)^{n'-m'},
\tag{A8c}
$$

$$
[\beta_4]_b = \frac{\exp\left[|\tilde{\beta}(t)|^2 - |\beta'|^2\right]\exp\left[|\beta'|^2\frac{N(t)}{N(t)+1}\frac{\bar{n}_{\text{th}}+1}{\bar{n}_{\text{th}}}\right]}{N(t)+1}\sum_{n'=0}^{\infty}\frac{1}{n'!}\left[\frac{N(t)}{N(t)+1}\right]^{n'}
$$
$$
\times\sum_{k'=0}^{n'}\sum_{m'=0}^{n'}\binom{n'}{k'}\binom{n'}{m'}\sqrt{k'!}\sqrt{m'!}\,\tilde{\beta}^*(t)^{n'-k'}\tilde{\beta}(t)^{n'-m'}.
\tag{A8d}
$$

In these expressions, the terms $\tilde{\alpha}(t) = \frac{\alpha' e^{-\gamma t}}{N(t)+1}$ and $\tilde{\beta}(t) = \frac{\beta' e^{-\gamma t}}{N(t)+1}$ represent the time-dependent coefficients that include the effects of the decay rate $\gamma$ and the factor $N(t)$, which depend on the thermal environment.

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
