# Peer review of "Dynamics of the Interaction Between Two Coherent States in a Cavity with Finite Temperature Decay"

_2673-8716, doi:10.3390/dynamics5010004_

Round 1
Reviewer 1 Report
Comments and Suggestions for Authors
The topic of the dynamics of the coherent states with the interaction between the system and its cavity is interesting, especially for the case of finite temperature decay. In this manuscript the authors investigate the dynamics via finding the exact solution of the corresponding Lindblad master equation.
The manuscript could be understood hardly, the results in the manuscript could be the existing formally. I give my concise comments as follows.
1 what is the physical meaning of the operator S? Does the operators a and b commute with each other?
2 the solution of the Lindblad equation of Eq. 1 can, even under the condition of eq. 3, NOT be written as eq. 4 since eq. 1 is not the usually operator action.
3 also, how is eq. 11 obtained from eq. 9 since the actually meaning of the operator action is eq. 10.
4 there is no physical example to present the authors’ results.
Author Response
Reviewer’s Comment: what is the physical meaning of the operator S? Do the operators a and b commute with each other?
Response:
The operator S represents the unitary evolution in the interaction picture, specifically describing the system’s evolution in the absence of decay. The operators a (a†) and b (b†) commute with each other, as they act on different modes. However, the interaction term ab† + a†b does not generally commute with the dissipative terms Laρ + Lbρ. Nevertheless, when identical decay rates are assumed, γ =γa = γb, the associated superoperators commute, as noted in the manuscript. To clarify this point, we have made this observation more explicit in the main text, immediately following Eq. (3). Furthermore, the caption of Fig. 1 has been revised to reflect this updated explanation and ensure consistency with the text.
2. Reviewer’s Comment: the solution of the Lindblad equation of Eq. 1 can, even under the condition of Eq. 3, NOT be written as Eq. 4 since Eq. 1 is not the usual operator action.
Response:
Eq. (1) represents the standard form of the Lindblad equation in the interaction picture. Under the condition that the decay rates for both reservoirs are identical, i.e., γ = γa = γb, and assuming that both reservoirs are characterized by the same thermal occupation number ¯nth, the formal and exact solution can indeed be written in the form of Eq. (4). This formulation is consistent with the condition described by Eq. (3) and follows from the symmetry of the reservoirs and the structure of the Lindblad operators. To address the reviewer’s concern, we have added a detailed explanation immediately after Eq. (3), marked in red, to clarify the assumptions and derivation leading to Eq. (4). Furthermore, we have revised the caption of
Fig. 1 to explicitly highlight these assumptions, ensuring the discussion is coherent, clear, and self-contained.
3. Reviewer’s Comment: also, how is Eq. 11 obtained from Eq. 9 since the actual meaning of the operator action is Eq. 10.
Response:
Thank you for pointing out this important aspect. To address this concern, we thoroughly revised Section 3 to provide a clearer and more detailed explanation of the derivation connecting Eqs. (9), (10), and (11). Specifically, we added intermediate steps and expanded the mathematical justification to explain how the operator action described in Eq. (10) leads to the form of Eq. (11). While the results themselves remain unchanged, the revised explanation ensures that the logical flow and the relationship between these equations are explicitly clear. We deeply appreciate these insightful comments, which allowed us to significantly improve the coherence and comprehensibility of this section.
4. Reviewer’s Comment: there is no physical example to present the authors’ results.
Response:
We sincerely thank you for pointing out this important aspect. To address this concern, we have made significant revisions to the manuscript, which include:
â—‹ Addition to Section IV: At the end of Section IV, we have introduced a new subsection where we present our numerical results. This subsection includes an analysis of the entropy to discuss the preservation or modification of entanglement under Lindblad dynamics, providing a clearer connection to physically meaningful quantities.
â—‹ New Section V: We have added a dedicated section to analyze a specific physical example for the case of an initially entangled state. In this section, we also present corresponding numerical results to complement the theoretical analysis and offer deeper insights into the dynamics.
These additions ensure that our results are not only well-founded theoretically but also illustrated through practical examples, making the manuscript more comprehensive and accessible to the readers. We are grateful for this valuable suggestion, which has significantly enhanced the overall quality and relevance of our work.
Reviewer 2 Report
Comments and Suggestions for Authors
Comments attached as a file.

Author Response
1. Reviewer’s Comment: One of the main conclusions stated by the authors is that if the system begins in an entangled coherent state, the dynamics preserves this entanglement even in the presence of finite temperature dissipation. However, unless I have missed something, this is not actually demonstrated in the manuscript. The authors show that the dynamics will not generate any entanglement from a non-entangled initial state, but this is not equivalent to showing that entanglement will be preserved from an initially entangled state.
Response:
Thank you for pointing out this significant omission in the original manuscript. Your comment not only highlighted a critical aspect that had not been addressed in the initial work, but also motivated us to study this case in greater depth. As a result, we realized that some of our initial claims regarding the dynamics of entangled states were incorrect and needed to be revised. To address this issue, we have added a new section (Section V) in which we analyze the case of an initially entangled coherent state. This analysis is complemented by comprehensive numerical simulations that validate the theoretical results and provide a clearer understanding of the evolution of entanglement. In particular, we evaluate entanglement using entropy as a measure to explicitly demonstrate how quantum
correlations evolve over time in the presence of thermal dissipation. These additions not only directly address your important concern, but were also instrumental in refining and strengthening our conclusions regarding the behavior of entangled states.
2. Reviewer’s Comment: I think it would significantly help convey the main message of the manuscript to include a numerical demonstration of the results. Perhaps something like a plot of an appropriate entanglement measure as a function of evolution time for different choices of initial state.
Response:
We thank you for this valuable suggestion, which has significantly improved the clarity and presentation of our results. In response to your comment, we have included numerical demonstrations in the revised manuscript to complement the analytical derivations and strengthen the overall message. Specifically:
â—‹ In Section IV, we present numerical results for the case of an initially non-entangled coherent state. These results include graphs of the entropy of the system as a function of evolution time under different thermal dissipation conditions, clearly illustrating the absence of entanglement generation.
â—‹ In the newly added Section V, we analyze the dynamics of an initially entangled coherent state. This section includes plots of entropy as a measure of entanglement, demonstrating how quantum correlations evolve over time under varying initial conditions.
3. Reviewer’s Comment: There is a missing section number in the first paragraph on page 2.
Response:
Thank you for noticing this issue. We have corrected the missing section number in the revised manuscript.
Reviewer 3 Report
Comments and Suggestions for Authors
In the present paper, the authors study the dissipation dynamics of two interacting quantized fields each coupled to a Markovian thermal reservoir with finite temperature. The authors aim to derive the analytical expression of the Lindblad master equation for the dissipation dynamics. The study in the manuscript is sound and of interest to some physical communities and the manuscript is overall clear. However, to be accepted for publication, the author must consider certain aspects of the presentation and broaden discussions and relationships with previous articles in the literature. For an overall improvement of the manuscript, I would suggest the author further address the following points:
1. The authors should bring some new physical insight to make the readers better understand the background and the motivation of the study. I believe that the authors missed some important references about the study in the introduction section. Thus, the introduction section should be further broadened.
2. I consider the authors should also describe the physical models in detail in the methods part and some relative references should also be cited. For example, The authors should give the Hamiltonian of the total system described in Fig. 1 rather than only give the dynamical equation in Eq. 1. Moreover, the discussion and physical meanings should be further broadened. It is a better way to present the complete models and results so that the readers can better understand the importance of the study.
I can recommend the paper to be published in Dynamics after the authors have considered these items.
Author Response
1. Reviewer’s Comment: The authors should bring some new physical insight to make the readers better understand the background and the motivation of the study. I believe that the authors missed some important references about the study in the introduction section. Thus, the introduction section should be further broadened.
Response:
Thank you for this valuable suggestion. In response, we have significantly revised and expanded the Introduction section to provide a more comprehensive background and motivation for the study. We have included additional references to key works and we have broadened the discussion to highlight the broader implications of our research in quantum optics and quantum technologies. Additionally, we have clarified the motivation for
analyzing both entangled and non-entangled initial states, linking it to the challenges of preserving quantum correlations in practical systems. These updates position our work more effectively within the existing literature and provide readers with a clearer understanding of its significance. All changes have been colored red in the revised manuscript for easy identification.
2. Reviewer’s Comment: I consider the authors should also describe the physical models in detail in the methods part and some relative references should also be cited. For example, the authors should give the Hamiltonian of the total system described in Fig. 1 rather than only give the dynamical equation in Eq. 1. Moreover, the discussion and physical meanings should be further broadened. It is a better way to present the complete models and results so that the readers can better understand the importance of the study.
Response:
Thank you for this thoughtful and constructive comment. In response, we have revised the methods section to include a detailed description of the physical model, including the Hamiltonian of the total system, which explicitly describes the interaction between the field modes and the thermal reservoir. This addition establishes a clearer connection between the physical setup and the dynamical equation presented in Eq. (1). Furthermore, we have improved the text to better explain the underlying physics and broadened the discussion to emphasize how the interaction terms and dissipation parameters affect the dynamics of the system. To enhance clarity, we also added a new section with numerical results that complement the analytical solutions, providing a more comprehensive understanding of the physical implications of our findings. We are sincerely grateful for your suggestions, which have greatly improved the quality and presentation of our work.
Round 2
Reviewer 1 Report
Comments and Suggestions for Authors
In the revision the authors clean my concerns, and it can be accepted to publish.
Author Response
We sincerely appreciate your positive feedback and for acknowledging that our revisions have addressed the concerns raised. We are pleased to see that the manuscript is now considered suitable for publication. Thank you for your valuable comments and suggestions that have significantly contributed to improving the quality of our work.
Reviewer 2 Report
Comments and Suggestions for Authors
Comments attached as a file.

Author Response
Please find attached the answers in the Reviewer2.pdf file.

Reviewer 3 Report
Comments and Suggestions for Authors
I believe the authors have adequately addressed all my comments. The revised manuscript is much clearer than the previous version and provides enough novelty and significant results to meet the acceptance criteria of Dynamics. Therefore, I recommend the publication of the manuscript in its present form.
Author Response
We sincerely thank you for your positive comments and for recognizing the clarity, novelty, and relevance of our revised manuscript. We greatly appreciate the time and effort dedicated to reviewing our work and are very grateful for the recommendation for publication in its current form.
Round 3
Reviewer 2 Report
Comments and Suggestions for Authors
In the revised manuscript the authors have addressed all of my comments and concerns. I particularly appreciate the new analysis of the quantum discord, which I believe substantially increases the interest and significance of the authors' analysis. I have no further comments and recommend publication of the manuscript in its current state.